

# Phases of scrambling in eigenstates

Tarek Anous[1*] and Julian Sonner[2]

**1** Department of Physics and Astronomy, University of British Columbia,
Vancouver, BC V6T 1Z1, Canada
**2** Department of Theoretical Physics, University of Geneva,
24 quai Ernest-Ansermet, 1211 Genève 4, Suisse

* tarek@phas.ubc.ca

## Abstract

We use the monodromy method to compute expectation values of an arbitrary number of light operators in finitely excited ("heavy") eigenstates of holographic 2D CFT. For eigenstates with scaling dimensions above the BTZ threshold, these behave thermally up to small corrections, with an effective temperature determined by the heavy state. Below the threshold we find oscillatory and not decaying behavior. As an application of these results we compute the expectation of the out-of-time order arrangement of four light operators in a heavy eigenstate, i.e. a six-point function. Above the threshold we find maximally scrambling behavior with Lyapunov exponent $2\pi T_{\text{eff}}$. Below threshold we find that the eigenstate OTOC shows persistent harmonic oscillations.


# 1 Introduction and summary

In recent years it has been recognised that black holes exhibit a strong form of chaos, most cleanly quantified by a *thermal* out-of-time-order four-point function (OTOC) [1]

$$F(t) := \langle W(t)V(0)W(t)V(0) \rangle_\beta . \tag{1}$$

If this four-point function is evaluated in a CFT with holographic dual, for example a sparse large$-c$ CFT$_2$ as will be the focus of this work, one finds that it contains an exponentially growing piece

$$F(t) = F_0 - \frac{K}{c} e^{\lambda_L(t-x)} + \cdots , \tag{2}$$

where $K$ is a constant that depends on the choice of operators $W$ and $V$ and the Lyapunov exponent $\lambda_L = 2\pi T$ takes on its maximal allowed value [2]. At the same time, detailed calculations [3–11] reveal that highly excited pure states in theories with holographic duals can act thermally: for simple enough operators they reproduce the expectation values in a suitable thermal ensemble at late times, up to small corrections,

A compelling scenario explaining the thermalization of simple operators in closed unitarily evolving quantum systems is the eigenstate thermalization hypothesis (ETH) [12,13], which we will review in more detail below. Succinctly, ETH states that finitely-excited eigenstates themselves carry information about the thermal ensemble. This implies that typical states made from random superpositions of eigenstates in a small energy window, also look thermal, up to corrections that are exponentially small in entropy, when probed by simple enough operators,

$$\langle \Psi | \mathcal{Q}(t) | \Psi \rangle \longrightarrow \langle \mathcal{Q}(t) \rangle_{\beta_\Psi} + \mathcal{O}\left(e^{-S/2}\right) . \tag{3}$$

In contrast, non-thermalising systems, such as many-body-localised phases (MBL), do not satisfy ETH. As we will see below, eigenstates below the BTZ threshold do not satisfy ETH, and behave in ways more reminiscent of non-thermalizing phases.

The operator whose expectation value we calculate to obtain $F(t)$ in (1) is simple at early times, but becomes increasingly complicated as the time-evolved $W(t) = e^{iHt}We^{-iHt}$ spreads to encompass a larger and larger fraction of the total system. ETH therefore does not imply that $F(t)$, computed in a typical pure state, will approximate the answer in the thermal ensemble.

As an example, take the the $n^{\text{th}}$ Rényi entropy, which can be thought of as an operator whose spatial extent covers a finite fraction of the total system size. Only the limit $n \to 1$, namely the entanglement entropy, behaves approximately thermally when evaluated in an energy eigenstate, as shown in [14–16]. For holographic (that is sparse large-$c$ CFT), our result adds the OTOC to the list of operators of finite spatial extent whose expectation value is nevertheless approximately thermal in an eigenstate. There is evidence [8] that this should be the case for theories with a holographic dual, at least up to the scrambling time $t_* \sim \log S$, where the definition of the scrambling exponent (2) is valid. *In this paper we show that in fact a stronger result holds, which implies the approximately thermal behavior of the four-point OTOC in typical states as a corollary.*

**Main result**

In this work we use the monodromy method to establish that sparse, large$-c$ 2D CFT satisfy

$$\boxed{\left\langle H, \bar{H} | \mathcal{Q}_1(t_1)\mathcal{Q}_2(t_2)\cdots\mathcal{Q}_{n_L}(t_{n_L}) | H, \bar{H} \right\rangle \propto \text{Tr}\left[ e^{-\beta H}\mathcal{Q}_1(t_1)\mathcal{Q}_2(t_2)\cdots\mathcal{Q}_{n_L}(t_{n_L}) \right] + \mathcal{O}\left(c^{-1}, e^{-c}\right),}$$
$$\tag{4}$$

where $|H, \bar{H}\rangle := O_{H,\bar{H}}(0)|0\rangle$ is a heavy primary state with dimension $H + \bar{H} = \Delta \sim \mathcal{O}(c)$ and the 'probe' operators $\mathcal{Q}$ all have dimension $\Delta \sim \mathcal{O}(\varepsilon c)$ in terms of the central charge, for $\varepsilon \ll 1$.

The inverse temperature $\beta$ appearing in the canonical density matrix on the right-hand side of (4) is the one associated to the eigenstate $|H\rangle$[1] by a naive application of ETH. The notation $\mathcal{O}\left(c^{-1}, e^{-c}\right)$ is meant to indicate that the leading order result receives both perturbative corrections, as well as non-perturbative corrections in the central charge, the latter corresponding to the appearance of heavy conformal families in intermediate channels. The precise form of these corrections reveals interesting connections with conserved higher KdV charges and the corresponding generalized Gibbs ensemble [17–20].

Choosing the light composite operator in (4) to be the out-of-time-order arrangement mentioned above then immediately implies the result that there exists a notion of scrambling exponent in eigenstates, which moreover satisfies

$$\lambda_L = 2\pi T_{\text{ETH}}, \tag{5}$$

saturating an eigenstate version of the fast-scrambling bound [2, 21], as conjectured in [8] in the context of the SYK model. For future reference, let us mention the convention that we refer to correlations of the type (4) above as "HHLL...L", which is meant to indicate that there are two (H)eavy insertions and a number, typically > 2, of (L)ight ones. Note that [4] described a purely algebraic method to obtain (4), relying on $1/c$ scalings of the Virasoro generators when computing blocks of the form HHLL...L, whereas we prove this statement using the monodromy method. An important open problem is to determine the timescale at which the identity block domination breaks down in a given correlation function, in any given CFT. A natural candidate in sparse large-$c$ theories is the scrambling time $t_*$ itself.

### A scrambling phase transition

As we have already remarked, the thermality result (4) is valid only for heavy enough states, specifically above the microcanonical BTZ threshold. For such states the system behaves effectively ergodically, the holographic interpretation being that these pure states act as if there were a bulk horizon causing the exponential Lyapunov behavior. However, below threshold this is not the case, and thus one might naturally expect the exponential Lyapunov growth to be absent. Below we show that indeed this is the case, and the scrambling four-point function transitions to an oscillatory behavior. We have just used our bulk intuition for this transition, which makes it very natural, but it is important to emphasize that it is very non-trivial from the boundary theory point of view: we have uncovered a sharp transition in chaotic behavior that signals that the many-body system under consideration goes from an ergodic phase to a non-ergodic phase. In the discussion section we comment further on our findings, and comment on its relation to other non-ergodic phases violating ETH, such as many-body localized (MBL) systems.

At finite temperature, decaying OTOCs arise due to the exchange of a reparametrization mode or 'scramblon' in the Regge correlator (see e.g. [22–25]). As depicted in figure 1, our results imply that a similar scramblon is responsible for the decay or oscillation of OTO correlators in eigenstates, depending on whether the eigenstate is above or below the BTZ threshold.

### Holographic bulk reconstruction and black holes

Statements of the kind (4) are intimately related to a number of recent developments surrounding the holographic understanding of black holes and their microstates and attempts to address the firewall paradox raised in [27]. For example the authors of [28] point out that if statements of the type (4) & (5) hold for typical states, then a typical microstate of a black

---

[1]In this work we will only consider scalar operators with $H = \bar{H}$, and thus for the most part suppress the anti-holomorphic labels. We expect the overall picture to apply also to spinning operators, with separate left- and right-moving effective temperatures. See appendix A.

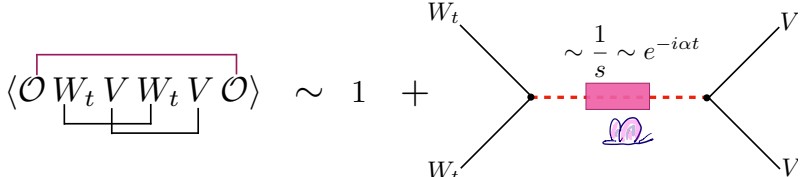

Figure 1: Application of our main result (4) to the butterfly effect ('scrambling') in heavy states, i.e. an out-of-time-order "HHLLLL" correlation function. As indicated in purple, the effect of the heavy insertions can be interpreted as dressing the propagator, $s^{-1}$, of the leading mode, the "scramblon" [26], exchanged between the light operators on the second sheet by the exponential factor $s^{-1} = e^{-i\alpha t}$. In the ergodic phase, $\alpha = i|\alpha|$, this leads to maximal scrambling, while in the non-ergodic phase, $\alpha = |\alpha|$, this results in an oscillatory OTOC. The bulk interpretation of the scramblon picture is a single graviton exchange in the heavy background. Full details of the calculation summarized here can be found in section 4 below.

hole itself contains information about behind-the-horizon physics. Our results hold for eigenstates and will thus extend to typical states that are superpositions of eigenstates in a small energy window as required by [28]. More precisely these authors show that if a statement of the form (4) holds, one can add a "double-trace" deformation $\mathcal{Q}\tilde{\mathcal{Q}}$ to the CFT, where $\tilde{\mathcal{Q}}$ is the so-called mirror operator of [29], which mimicks the double-trace deformation of Gao, Jafferis and Wall [30], and thereby extract information about the region behind the horizon in a typical microstate.

Very recently, [31] (building on [32, 33]) similarly proposed a link between the firewall paradox and efficient scrambling, albeit *not* in eigenstates. They demonstrate that, at infinite temperature, the decay of OTOCs implies a growing mutual information between the black hole interior and exterior as a function of time. This in turn implies that the Hayden-Preskill protocol [34] can be used to recover black hole information.

## 2 Background

Since we will frequently refer to the Eigenstate Thermalization Hypothesis in what follows, we will briefly introduce and review it. The ETH can be stated relatively simply. Let us consider a non-extensive operator $\mathcal{Q}$ in a system that obeys ETH. This means that $\mathcal{Q}$'s matrix elements between energy eigenstates $\{|n\rangle\}$ must take the following form:

$$\langle m|\mathcal{Q}|n\rangle = \overline{\mathcal{Q}}(\overline{E})\,\delta_{mn} + e^{-S(\overline{E})/2} f\left(\overline{E}, \omega\right) R_{mn} \, , \tag{6}$$

where $\overline{\mathcal{Q}}(\overline{E})$ is the microcanonical average of the operator $\mathcal{Q}$, that is, over a set of energy eigenstates in a small band centered around $\overline{E}$, with uniform coefficients. $R_{mn}$ is a random variable with zero mean and unit variance, and the only restriction on $f$ is that it must be a smooth function of $\overline{E}$ and $\omega := E_n - E_m$. If (6) is satisfied, then the expectation value of $\mathcal{O}$ in a *single* eigenstate of energy $\overline{E}$ is the same as its microcanonical average, up to exponentially suppressed terms.

The equivalence between microcanonical and canonical averages implies a relationship between $\overline{E}$, the center of the microcanonical enegy band, and an "effective temperature" $1/\beta$, given by

$$\partial_E S(E) = \beta \, , \tag{7}$$

where $S(E)$ is the logarithm of the number of states at energy $E$.

The entropy of two-dimensional CFTs quantized on a spatial circle and at large central charge is fixed, at high energies, by modular invariance (as well as certain additional assumptions [35–37]). For spinless operators of weight $(H, H)$ it takes the form:

$$S(E) = 2\pi\sqrt{\frac{c}{3}E}\,, \qquad E \equiv 2H - \frac{c}{12}\,. \tag{8}$$

This, in turn, fixes the microcanonical inverse temperature:

$$\beta_H \equiv \frac{2\pi}{\sqrt{\frac{24H}{c} - 1}}\,. \tag{9}$$

The inverse temperature (9) has appeared numerous times in recent years in showing how certain CFT pure states reproduce the physics outside of black hole backgrounds in AdS$_3$ [3, 4, 38, 39]. In this paper we add to that list by showing how (9) appears in a calculation of the Lyapunov exponent computed in a single CFT eigenstate, and in fact the more general result (4). As mentioned in the introduction, and modulo certain reasonable assumptions, we show that the Lyapunov exponent is maximal and is given by $\lambda_L = \frac{2\pi}{\beta_H}$. On the one hand, the fact that $\lambda_L$ is maximal in CFT pure states is expected if the theory has a holographic dual and obeys ETH. On the other hand, it is interesting to see precisely how one obtains this result. Doing so sheds light on how ETH-like calculations pan out in 2d CFTs at large central charge. To wit, our proof requires expanding upon techniques developed in [4], and in doing so we clarify some statements made about CFT pure states reproducing thermal answers (see specifically section 5.1 of [4]).

## 2.1 Our approach

Let us now expand on our approach to the microstate average in 2D CFT. By the operator-state correspondence we can map the computation of $n_L$ light operators in the state created by a heavy operator to a normalized $n_L + 2$ correlation function, so that the statement (4) takes the form

$$\frac{\langle O_H(\infty)\,\mathcal{Q}_1(z_1)\,\mathcal{Q}_2(z_2)\dots\,\mathcal{Q}_{n_L}(z_{n_L})\,O_H(0)\rangle}{\langle O_H(\infty)O_H(0)\rangle} = \langle \mathcal{Q}_1(x_1)\,\mathcal{Q}_2(x_2)\dots\,\mathcal{Q}_{n_L}(x_{n_L})\rangle_{\beta_H} + \mathcal{O}\left(c^{-1}, e^{-c}\right), \tag{10}$$

for the CFT quantized on a spatial circle. The usual operator-state map computes the expectation value with respect to the heavy eigenstates on the circle. This explains our choice of coordinates $z_i = e^{ix_i}$. $O_H$ and $\mathcal{Q}_i$ are scalar primary operators, and we have suppressed the dependence on anti-holomorphic variables. The insertion of the heavy operator at the origin creates the ket $|H\rangle$ in the infinite past on cylinder, while the insertion at infinity creates the conjugate bra $\langle H|$. Thus (10) expresses once more that the expectation value of the $\mathcal{Q}_i$ in the state created by inserting $O_H$ at the origin match with thermal expectation values at the inverse temperature $\beta_H$, up to $1/c$ corrections. Our proof will hold so long as there is a separation of scales between the conformal dimensions of the $\mathcal{Q}_i$ and $O_H$, that is $H \gg h_{\mathcal{Q}_i}$ and in the regime where the identity block dominates. This block can certainly be made to dominate for larger separations by tuning the density of states such that the spectrum is sufficiently sparse [3, 6, 37, 40–46].

However, it is important to note that for the purposes of studying the chaotic dynamics, further constraints on the spectrum are required such that the identity block dominates in the Regge limit (required for extracting the chaos exponent). This has been investigated in [47–49] although a full characterization of what is required such that the identity block dominates on the second sheet is not yet fully understood.

What will naturally come out of this proof is that (10) matches thermal correlators evaluated for CFTs quantized on a spatial line, rather than a circle, and only holds for $H > c/24$ [3, 4, 42, 50, 51]. One may take this as evidence that these results only hold in the high temperature limit ($H \gg c/24$). However, when comparing to holographic calculations done in the BTZ geometry with spherical boundary, one finds that the free energy [52] is extensive in the size of the dual CFT and the entanglement entropy [53] is insensitive to the size of the CFT sphere. Both of these holographic results mimic thermal physics on the line rather than the circle. Thus holography requires that the large-$c$ limit behave essentially like a large volume limit for certain observables. One unifying picture for this behavior is that of unbroken center symmetry, which guarantees the volume independence of the free energy and entanglement entropy [54].

Since our proof will rely on the monodromy method [55], it therefore only holds at large central charge $c$ in the limit where $H/c$ and $h_{\mathcal{Q}_i}/c$ are held fixed. This is precisely the holographic limit where we expect to match with bulk geodesic calculations.

## 2.2 Thermal Jacobians

To motivate our result we will use two facts. The first is that primary operators of dimension $(h, \bar{h})$ transform under coordinate transformations $z \to z(w)$ as follows:

$$O(z, \bar{z}) \to \left( \frac{dz}{dw} \right)^h \left( \frac{d\bar{z}}{d\bar{w}} \right)^{\bar{h}} O(z(w), \bar{z}(\bar{w})) \; . \tag{11}$$

The second fact we will use is that thermal correlation functions (on the line) can be obtained from vacuum correlation functions using the conformal map from the plane to the cylinder $z \to e^{\frac{2\pi}{\beta} w}$. That is:

$$\left( \prod_i \left( \frac{dz_i}{dw_i} \right)^{h_i} \left( \frac{d\bar{z}_i}{d\bar{w}_i} \right)^{\bar{h}_i} \right) \left\langle \prod_i O_i (z_i(w_i), \bar{z}_i(\bar{w}_i)) \right\rangle = \left\langle \prod_i O_i (w_i, \bar{w}_i) \right\rangle_\beta \; , \tag{12}$$

where the expectation value on the left hand side is taken in the CFT vacuum, and the expectation value on the right hand side is taken in the thermal state with inverse temperature $\beta$. This leads to an important realization: in order for an eigenstate expectation value to act approximately thermally, the eigenstate must effectively enact the above coordinate transformation in (12), including the Jacobian term, with respect to *some emergent temperature* that should depend on the eigenstate in a way analogous to the original ETH.

It is important for the arguments we present below that thermal correlators, again on the line, can be obtained from vacuum correlators via this simple coordinate transformation. Thus, in order to obey ETH, we must show in what sense eigenstates behave like simple coordinate transformations *with no additional features*. This is crucial and, we will see that this only holds in a particular identity block approximation to the correlation function in the heavy eigenstate.

# 3 Proof of statement (4)

## 3.1 Monodromy basics

A few assumptions about correlation functions in 2d CFT go into this proof. Firstly, we assume that the correlation functions can be decomposed into Virasoro conformal blocks

$$G(z_i, \bar{z}_i) = \sum_k a_k \mathcal{F}_k(z_i) \bar{\mathcal{F}}_{\bar{k}}(\bar{z}_i) \; , \tag{13}$$

where the index $k$ labels which Virasoro primaries run in the OPE and the $a_k$ schematically represent products over OPE coefficients. We note that each individual conformal block $\mathcal{F}$ ($\bar{\mathcal{F}}$) depends (anti-)holomorphically on the locations of the insertions. We will also use the fact that at large-$c$, the blocks exponentiate :

$$\mathcal{F}_k(z_i) \sim e^{-\frac{c}{6}f_k(z_i)} \,. \tag{14}$$

The $f_k$ are often called semi-classical blocks and can be computed using the monodromy method.

The monodromy method, used for computing semiclassical conformal blocks, is explained in many places (see e.g. [41]), and here we give a cursory review of the basic ingredients required to follow the proof. Consider the holomorphic differential equation

$$\psi''(z) + T_{\text{cl}}(z)\psi(z) = 0 \,, \tag{15}$$

where $T_{\text{cl}}$ is the stress tensor expectation value arising from the insertion of our CFT operators. By the conformal Ward identity it must take the form[2]

$$T_{\text{cl}} = \sum_{k=1}^{n_H}\left(\frac{6H_k/c}{(z-y_k)^2} - \frac{b_k}{z-y_k}\right) + \sum_{i=1}^{n_L}\left(\frac{6h_i/c}{(z-z_i)^2} - \frac{c_i}{z-z_i}\right) \,, \tag{16}$$

where the $(b_i, c_i)$, called *accessory parameters*, are undetermined functions of the insertion points $(y_i, z_i)$. In (16) we have split the stress tensor into contributions from *heavy insertions* with coordinates labeled $y_i$ and accessory parameters labeled $b_i$, and *light insertions* with coordinates labeled $z_i$ and accessory parameters labeled $c_i$. Regularity at the origin under a coordinate inversion fixes the stress tensor to fall off as

$$T_{\text{cl}}(z \to \infty) = \mathcal{O}\left(z^{-4}\right) \,, \tag{17}$$

which imposes the following conditions on the accessory parameters:

$$\sum_{i=1}^{n_L} c_i = -\sum_{i=1}^{n_H} b_i \,, \tag{18}$$

$$\sum_{i=1}^{n_L}\left(c_i z_i - \frac{6h_i}{c}\right) = -\sum_{i=1}^{n_H}\left(b_i y_i - \frac{6H_i}{c}\right) \,, \tag{19}$$

$$\sum_{i=1}^{n_L}\left(c_i z_i^2 - \frac{12h_i}{c}z_i\right) = -\sum_{i=1}^{n_H}\left(b_i y_i^2 - \frac{12H_i}{c}y_i\right) \,. \tag{20}$$

To fix the remaining accessory parameters, we tune the $(b_i, c_i)$ such that the solutions to (15) obey certain monodromy conditions. For example, say that we wish to compute the block corresponding to the OPE channel: $O_A O_B \to O_C$, then we demand that the two independent solutions $\psi_{1,2}$ to (15) obey the following monodromy condition around a path $\gamma$ encircling both $z_A$ and $z_B$:

$$\begin{pmatrix}\psi_1 \\ \psi_2\end{pmatrix} \to M_\gamma \begin{pmatrix}\psi_1 \\ \psi_2\end{pmatrix} \,, \qquad \text{Tr}\, M_\gamma = -2\cos\left(\pi\sqrt{1 - \frac{24h_C}{c}}\right) \,. \tag{21}$$

Once we have fixed the $c_i$, the corresponding semiclassical block is obtained by solving

$$\frac{\partial f_k}{\partial y_i} = b_i \,, \qquad \frac{\partial f_k}{\partial z_i} = c_i \,. \tag{22}$$

A special mention is reserved for the semiclassical identity block, corresponding to $\text{Tr}\, M_\gamma = 2$, which contributes in all CFTs and appears to give universal results reproducing semiclassical gravity in AdS$_3$ [3, 4, 6, 42, 44].

---

[2]As is conventional, we have multiplied the stress tensor by $6/c$ for convenience.

## 3.2 Coordinate transformation

For the remainder of the paper we will take $n_H = 2$ and set $H_1 = H_2 = H$. We can now fix the $b_i$ as well as $c_1$ using (18), giving[3]

$$
\frac{c}{6} T_{\rm cl} = \frac{H(y_1 - y_2)^2}{(z - y_1)^2 (z - y_2)^2} +
$$
$$
\sum_{i=1}^{n_L} \left[ h_i \left( \frac{1}{(z - z_i)^2} - \frac{(z - z_1) + (z_i - y_1) + (z_i - y_2)}{(z - z_1)(z - y_1)(z - y_2)} \right) - \frac{c}{6} \frac{c_i (z_i - z_1)(z_i - y_1)(z_i - y_2)}{(z - z_1)(z - z_i)(z - y_1)(z - y_2)} \right] .
\tag{23}
$$

It is known that under local coordinate transformations $z \to w(z)$, the stress tensor transforms as a quasi-primary of weight 2:

$$
T_{\rm cl}(z) = w'(z)^2 \, \tilde{T}_{\rm cl}(w(z)) + \frac{1}{2} \{ w(z), z \} , \qquad \{ w(z), z \} = \frac{w'''(z)}{w'(z)} - \frac{3}{2} \left( \frac{w''(z)}{w'(z)} \right)^2
\tag{24}
$$

and $\psi(z)$ transforms as a weight $-1/2$ density

$$
\psi(z) \to w'(z)^{-1/2} \, \psi(w(z)) .
\tag{25}
$$

Applying these transformations to (15) appears to transform it trivially:

$$
w'(z)^{3/2} \left[ \frac{d^2}{dw^2} \psi(w(z)) + \tilde{T}_{\rm cl}(w(z)) \psi(w(z)) \right] = 0 .
\tag{26}
$$

But that is not quite correct, the advantage we have gained is that we can reinterpret the new differential equation from the intrinsic geometry of the $w$ plane. Taking $w$ as the fundamental coordinate, we have:

$$
\psi''(w) + \left[ z'(w)^2 \, T_{\rm cl}(z(w)) + \frac{1}{2} \{ z(w), w \} \right] \psi(w) = 0 .
\tag{27}
$$

As was first noted in [4], choosing

$$
z(w) = y_2 \frac{w^{1/\alpha} + y_1}{w^{1/\alpha} + y_2} \qquad \text{with} \qquad \alpha = \sqrt{1 - \frac{24H}{c}}
\tag{28}
$$

eliminates the terms in $T_{\rm cl}$ proportional to $H$. In this sense, the $w$ coordinate would seem to trivialize their effect.

We are now tasked with understanding the monodromy properties of the following differential equation:

$$
\psi''(w) + \tilde{T}_{\rm cl}(w) \psi(w) = 0 ,
\tag{29}
$$

with

$$
\frac{c}{6} \tilde{T}_{\rm cl}(w) = \frac{w^{\frac{1 - 2\alpha}{\alpha}}}{\alpha^2} \times
$$
$$
\sum_{i=1}^{n_L} \left[ h_i \frac{w_i^{1/\alpha} \left( w_1^{1/\alpha} - w_i^{1/\alpha} \right)^2 - y_2 \left( w^{1/\alpha} - w_i^{1/\alpha} \right)^2 + \left( y_2 w^{1/\alpha} + w_1^{1/\alpha} w_i^{1/\alpha} \right) \left( w^{1/\alpha} - w_1^{1/\alpha} \right)}{\left( w^{1/\alpha} - w_i^{1/\alpha} \right)^2 \left( w^{1/\alpha} - w_1^{1/\alpha} \right) \left( w_i^{1/\alpha} - y_2 \right)} \right.
$$
$$
\left. - \frac{c}{6} \frac{c_i \, w_i^{1/\alpha} y_2 \left( w_i^{1/\alpha} - w_1^{1/\alpha} \right) (y_2 - y_1)}{\left( w^{1/\alpha} - w_i^{1/\alpha} \right) \left( w^{1/\alpha} - w_1^{1/\alpha} \right) \left( w_i^{1/\alpha} + y_2 \right)^2} \right] .
\tag{30}
$$

---

[3]Normal treatments take $(y_1, y_2, z_1) \to (0, \infty, 1)$ but we will keep them arbitrary here.

While we may have gotten rid of two insertions, this new stress tensor looks terrible! But note that we only care about the monodromy properties of (29), meaning we can simply study (30) near its singular points. Up to regular terms as $w \to w_i$, we find:[4]

$$\frac{c}{6} \tilde{T}_{\text{cl}} = \frac{h_1}{(w-w_1)^2} - \frac{h_1 + \sum_{i=2}^{n_L}\left(h_i - \frac{c}{6}\tilde{c}_i w_i\right)}{w_1(w-w_1)} + \sum_{i=2}^{n_L}\left[\frac{h_i}{(w-w_i)^2} - \frac{c}{6}\frac{\tilde{c}_i}{w-wi}\right] + \text{reg.}, \quad (31)$$

where we have defined

$$\tilde{c}_i := c_i z'(w_i) - \frac{6h_i}{c}\frac{\partial}{\partial w_i}\log z'(w_i), \qquad z(w) = y_2\frac{w^{1/\alpha} + y_1}{w^{1/\alpha} + y_2}. \quad (32)$$

It would appear that the $w$-plane monodromy problem is simply a standard monodromy problem with a newly defined set of accessory parameters. But this is too hasty. In fact we have a *non-standard* monodromy problem, since we can no longer fix the $\tilde{c}_i$ by demanding regularity at infinity on the $w$-plane. *Regularity does not strike twice.* To further elucidate this: demanding regularity as $w \to \infty$ would be equivalent to asking for regularity at $z = y_2$, which would be incorrect since we have inserted an operator there. In fact, the new stress tensor behaves as:

$$\frac{c}{6}\tilde{T}_{\text{cl}}(w \to \infty) = -\frac{\sum_{i=1}^{n_L}\left(h_i - \frac{c}{6}\tilde{c}_i(w_i - w_1)\right)}{w_1 w}$$
$$+ \frac{\sum_{i=1}^{n_L}\left\{h_i(2w_i - w_1) - \frac{c}{6}\tilde{c}_i(w_i - w_1)w_i\right\}}{w^3} + \mathcal{O}\left(w^{-4}\right). \quad (33)$$

As we can see, $\tilde{T}_{\text{cl}}$'s behavior as $w \to \infty$ contains information about the block we're computing, through its dependence on the $\tilde{c}_i$.

So as a recap: we have a non-standard $w$-plane monodromy problem, since $\tilde{T}_{\text{cl}}$'s behavior as $w \to \infty$ is not fixed to fall off like $w^{-4}$ by regularity at the origin.

Now, recall that the semiclassical block is related to the accessory parameters by solving $\partial f_k / \partial z_i = c_i$ as in (22). Using this we can reinterpret (32):

$$\frac{\partial \tilde{f}_k}{\partial w_i} = \frac{\partial f_k}{\partial z_i}\frac{\partial z_i}{\partial w_i} - \frac{6h_i}{c}\frac{\partial}{\partial w_i}\log z'_i(w_i), \quad (34)$$

which up to the logarithmic term is simply an application of the chain rule. What does this imply for the individual blocks $\mathcal{F}_k$? We can reinterpret (34) in the original coordinates $z_i$:

$$\frac{\partial f_k}{\partial z_i} = \frac{\partial \tilde{f}_k}{\partial w_i}\frac{\partial w_i}{\partial z_i} - \frac{6h_i}{c}\frac{\partial}{\partial z_i}\log w'_i(z_i), \quad (35)$$

which means that, solving the monodromy problem in the $w_i$ coordinates allows us to immediately find the leading contribution to the block in the $z_i$ coordinates

$$\boxed{\mathcal{F}_k(z_i) \approx e^{-\frac{c}{6}f_k(z_i)} = e^{-\frac{c}{6}\tilde{f}_k(w_i(z_i))}\prod_{i=1}^{n_L}(w'_i(z_i))^{h_i}.} \quad (36)$$

In general there is an integration constant in going from (35) to (36), but for the special case of the identity block it vanishes, unlike non-identity blocks (see appendix B). We will

---

[4]It is important that we drop the regular terms actually, as this is the invariant information carried by the Virasoro symmetries. In fact the form in (31) is required by the Virasoro Ward identity in the $w$ coordinate.

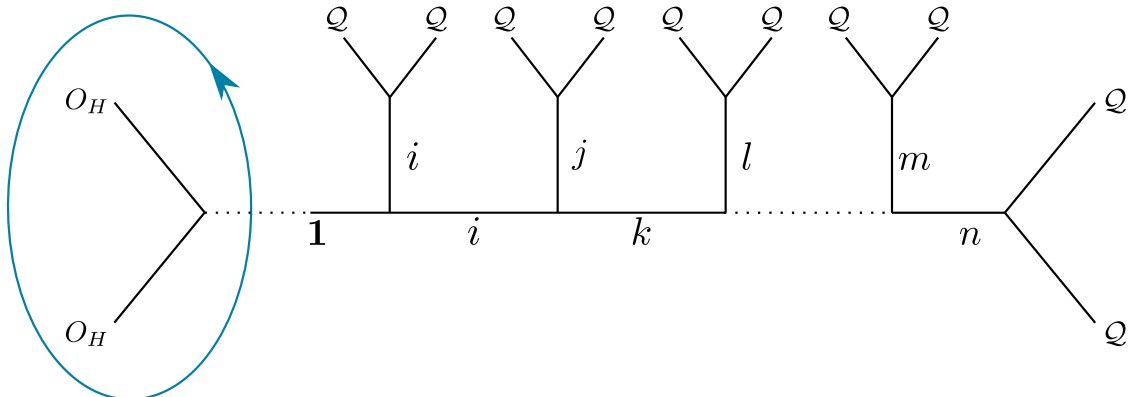

Figure 2: We will treat the $\mathcal{Q}$ operators as parametrically lighter than the $O_H$. We can then impose that the $O_H$ fuse to the identity operator, which, to leading order results in $\tilde{T}_{\rm cl}(w \to \infty) = O(w^{-4})$.

address the caveats related to this expression shortly, but the message is as follows: if we can solve the $w$-plane problem, then we can extract the solution to the $z$-plane problem by undoing the coordinate transformation (28). This is precisely what we had hoped for given the motivation in section 2.2.

Before moving on, it is important to reiterate that (36) will only actually hold in the identity block, due to additional integration constants appearing in blocks where a non-identity operator is exchanged between the heavy and the light insertions (see appendix B).

### Some parameter counting

We started with a monodromy problem on the $z$-plane with $n \equiv n_H + n_L$ insertions and $n$ accessory parameters. Three of these are fixed by demanding regularity on the $z$ plane as in (18), leaving $n-3$ accessory parameters which must be tuned by imposing the proper monodromy conditions.

On the $w$-plane, we have $n_L$ insertions and $n_L - 1$ accessory parameters to be tuned by imposing the desired monodromies. The fact that we have $n_L - 1$, and not $n_L - 3$, accessory parameters is an artifact of the initial setup on the $z$-plane. However if we could somehow impose two additional conditions, we would cut down the number of accessory parameters to $n_L - 3$, thus returning us to the standard setup on the $w$-plane.

We need look no further than (33) to find these two extra conditions. If we could argue for regularity as $w \to \infty$, instead of the more general condition written there, we would be back to the standard case that we know and love.

### Heavy-light imit

So far nothing we have done assumed that the two operators of dimension $H$ are much heavier than the rest. But this is precisely the limit we need to take. If we treat the $n_L$ light insertions as a perturbation, then to leading order, the two insertions of dimension $H$ fuse to the identity. To a first approximation, this implies regularity as $w \to \infty$ or $z \to z_2$. This is precisely the limit we take in the rest of the paper as shown in figure 2.

### 3.3 Sanity check (HHLL done doubly well)

The simplest place to demonstrate this technique is the HHLL vacuum block, first computed in [3]. We start with the monodromy differential equation (29) with $\tilde{T}_{\text{cl}}$ given in (31). In this example $n_L = 2$ and we will also take $h_1 = h_2 = h_L$.

We are computing a two-point function in the $w$ plane, and the stress tensor $\tilde{T}_{\text{cl}}$ has one accessory parameter $\tilde{c}_2$. Fixing $\tilde{c}_2$ such that $\tilde{T}_{\text{cl}}(w \to \infty) \sim \mathcal{O}(w^{-4})$ yields:

$$\tilde{c}_2 = -\frac{12 h_L/c}{w_1 - w_2} \, , \tag{37}$$

which we can integrate to obtain the semiclassical block on the $w$ plane:

$$\tilde{f}_0(w_1, w_2) = \frac{12 h_L}{c} \log(w_1 - w_2) + \text{const} \, . \tag{38}$$

We now use (36) to translate this to the semiclassical block in the $z$ plane. Recall that the coordinates are related by

$$w(z) = \left( y_2 \frac{z - y_1}{y_2 - z} \right)^\alpha \, . \tag{39}$$

To fix the constant in (38), we require the correlator exhibit the correct $z$-plane singularity as $y_1 \to y_2$:

$$
\begin{aligned}
f_0(z) &= \frac{12 H}{c} \log(y_1 - y_2) + \frac{12 h_L}{c} \log(w(z_1) - w(z_2)) - \frac{6 h_L}{c} \left( \log w'(z_1) + \log w'(z_2) \right) \\
&= \frac{12 H}{c} \log(y_1 - y_2) + \frac{12 h_L}{c} \log\left[ (z_1 - z_2) \frac{z^{\frac{1-\alpha}{2}} (1 - z^\alpha)}{\alpha(1 - z)} \right] ,
\end{aligned}
\tag{40}
$$

where in (40) we have introduced the cross ratio $z$:

$$z \equiv \frac{(z_1 - y_2)(z_2 - y_1)}{(z_1 - y_1)(z_2 - y_2)} \, . \tag{41}$$

Equation (40) is precisely the answer found in [3] for the vacuum Virasoro block, which we have managed to compute without solving a monodromy problem. The addition of the Jacobian term arising from the coordinate transformation (39) was crucial in getting the correct answer. Notice, however, that fixing the stress tensor to vanish like $w^{-4}$ in turn completely fixes the accessory parameter, meaning this simple trick cannot work if we want to compute the block corresponding to non-identity exchange. We explain how to obtain the non identity blocks (still in the heavy light limit) in appendix B. It is important to mention that, in deriving the *non-identity* blocks using the $w$-plane geometry, we encounter an additional subtlety not present in the identity block calculation on the $w$-plane: we need to include additional integration constants in order to ensure the correct OPE singularities on the $z$-plane. This explains why non-identity blocks fail to exhibit thermal behavior.

### A comment on ETH

We will briefly review the standard observation that a two point function computed in the eigenstate created by $O_H$ behaves thermally. We want to compute:

$$\frac{\langle O_H(\infty) \mathcal{Q}(z_1) \mathcal{Q}(z_2) O_H(0) \rangle}{\langle O_H(\infty) O_H(0) \rangle} \equiv \frac{\langle H | \mathcal{Q}(z_1) \mathcal{Q}(z_2) | H \rangle}{\langle H | H \rangle} \, . \tag{42}$$

For sufficiently small separations $|z_1 - z_2|$ the vacuum block contribution (40) to the above correlator will dominate.

If we put our light operators along the unit circle, $z_i = e^{ix_i}$, then performing the coordinate transformation from $z \to x$ in the vacuum block gives

$$\lim_{\substack{y_1 \to 0 \\ y_2 \to \infty}} \frac{(z'(x_1))^{h_L}(z'(x_2))^{h_L} e^{-\frac{c}{6}f_0(z(x_i))}}{(y_1 - y_2)^{-2H}} = \left[\frac{2}{\alpha}\sin\left(\frac{\alpha(x_1 - x_2)}{2}\right)\right]^{-2h_L} , \qquad \alpha = \sqrt{1 - \frac{24H}{c}} .$$
(43)

Now if $24H/c > 1$ we can identify an effective temperature $\beta_H = \frac{2\pi}{\sqrt{\frac{24H}{c} - 1}}$ such that the holomorphic part of the correlator is the usual thermal answer:

$$\mathcal{F}_0(x) \approx \left[\frac{\beta_H}{\pi}\sinh\left(\frac{\pi(x_1 - x_2)}{\beta_H}\right)\right]^{-2h_L} ,$$
(44)

with the analogous expression for the antiholomorphic contribution. This effective temperature is expected from the microcanonical ensemble for a CFT quantized on a spatial $S^1$, as explained at the beginning of this paper, and, in addition to the two-point function described in this example, will be present in higher point functions as well.

Of course this cannot be the full answer, as this block has certain 'forbidden singularities' whenever the $x_i$ are separated by an integer muliple of $i\beta_H$. This observation is intricately linked with the information loss problem in AdS$_3$ [4, 6, 9, 42, 43, 56]. Also it is important to recall our comment from above that non-identity blocks do *not* behave thermally in the sense of (44).

**Remarks**

As we have shown, if we restrict to the identity block in the limit where the $n_L$ $\mathcal{Q}$-operators are parametrically light, we can obtain the identity block of two heavy operators and $n_L$ light operators by studying a standard monodromy problem involving only the $n_L$ light operators. For $n_L \geq 4$ this remains a very difficult problem. However, we may obtain the even higher point blocks, if we are willing to restrict to identity exchanges in all internal channels, by recursively applying the method described herein.

# 4 Lyapunov exponent in a heavy eigenstate

We now set out to solve a $w$-plane monodromy problem with $n_L=4$. We will take $h_1 = h_2 = h_V$ and $h_3 = h_4 = h_W$.

This $w$-plane monodromy problem has three accessory parameters, two of which we again fix by demanding that $\tilde{T}_{\text{cl}}$ fall off like $w^{-4}$ at infinity, since we are interested in the HHLLLL identity block. We are now left with the task of computing an LLLL identity block on the $w$ plane. The answer has been computed by algebraic means in appendix B of [3], but can also be reproduced using the monodromy method. We simply copy the answer here:

$$\tilde{f}_0(w_i) = \frac{12h_V}{c}\log(w_1 - w_2) + \frac{12h_W}{c}\log(w_3 - w_4) - \frac{12h_V h_W}{c^2}(1-w)^2 {}_2F_1(2,2,4,1-w), \quad (45)$$

where the cross ratio $w$ is defined as

$$w \equiv \frac{(w_1 - w_4)(w_2 - w_3)}{(w_1 - w_3)(w_2 - w_4)} .$$
(46)

We see that the LLLL identity block on the $w$-plane is the exponentiated single graviton global block. We can now go from this expression to the HHLLLL identity block on the $z$ plane by

implementing the coordinate transformation:

$$f_0(z_i) = \tilde{f}_0(w(z_i)) - \frac{6}{c}\Big[h_V \log w'(z_1) + h_V \log w'(z_2) + h_W \log w'(z_3) + h_W \log w'(z_4)\Big] + \text{const.}$$

The coordinate transformation back to the $z$-plane is given in (39). Again we fix the constant by demanding the correct OPE singularity on the $z$-plane when $y_1 \to y_2$. Once the dust settles, we are left with the following semiclassical block

$$
\begin{aligned}
f_0 = {}&\frac{12H}{c}\log(y_1 - y_2) + \frac{12h_V}{c}\log\left[(z_1 - z_2)\frac{z^{\frac{1-\alpha}{2}}(1-z^\alpha)}{\alpha(1-z)}\right] \\
&+ \frac{12h_W}{c}\log\left[(z_3 - z_4)\frac{(u/v)^{\frac{1-\alpha}{2}}(1-(u/v)^\alpha)}{\alpha(1-(u/v))}\right] \\
&- \frac{12h_V h_W}{c^2}\left(1 - \frac{(1-u^\alpha)(z^\alpha - v^\alpha)}{(1-v^\alpha)(z^\alpha - u^\alpha)}\right)^2 {}_2F_1\left(2,2,4,1-\frac{(1-u^\alpha)(z^\alpha - v^\alpha)}{(1-v^\alpha)(z^\alpha - u^\alpha)}\right),
\end{aligned}
\tag{47}
$$

where we have introduced the following cross ratios:

$$z \equiv \frac{(z_1 - y_2)(z_2 - y_1)}{(z_1 - y_1)(z_2 - y_2)}, \quad u \equiv \frac{(z_1 - y_2)(z_3 - y_1)}{(z_1 - y_1)(z_3 - y_2)}, \quad v \equiv \frac{(z_1 - y_2)(z_4 - y_1)}{(z_1 - y_1)(z_4 - y_2)}. \tag{48}$$

In (47) we see two HHLL identity blocks, as well as a new piece that comes from enacting the coordinate transformation on the single graviton global block. We now proceed to use this new semiclassical identity block to extract some interesting physics.

## 4.1 Extracting the chaos exponent

Using (47) we can finally arrive at our main result. We will show that the Lyapunov exponent, as extracted from an out-of-time-ordered correlator computed in a heavy eigenstate, saturates the microcanonical chaos bound. That is $\lambda_L = 2\pi/\beta_H$ with $\beta_H = \frac{2\pi}{\sqrt{\frac{24H}{c}-1}}$. To extract the chaos exponent in the eigenstate $H$ we must compute a correlator of the form:

$$\text{OTOC} \equiv \frac{\langle H|V(t)W(x)V(t)W(x)|H\rangle}{\langle H|H\rangle}\frac{\langle H|H\rangle^2}{\langle H|V(t)V(t)|H\rangle\langle H|W(x)W(x)|H\rangle}. \tag{49}$$

We have divided out by the partially disconnected 4-point contributions and normalized each correlator in the eigenstate $|H\rangle$. We will use (47) for the 6-point contribution, while the factors in the denominator will be approximated using the standard HHLL identity block (40).

Since we want to compute these correlation functions in an eigenstate created by the insertion of $O_H$, we take $y_1 \to 0$ and $y_2 \to \infty$. Now, for the purposes of extracting the OTOC, we only need the leading order contribution in an expansion in small $h_V h_W/c$. In this limit, the holomorphic dependence of the identity block is simply:

$$\frac{\langle H|V(z_1)V(z_2)W(z_3)W(z_4)|H\rangle\langle H|H\rangle}{\langle H|V(z_1)V(z_2)|H\rangle\langle H|W(z_3)W(z_4)|H\rangle} \approx 1 + \frac{2h_V h_W}{c}s^2{}_2F_1(2,2,4,s) + \dots, \tag{50}$$

where

$$s \equiv \frac{\left(z_1^\alpha - z_2^\alpha\right)\left(z_3^\alpha - z_4^\alpha\right)}{\left(z_3^\alpha - z_2^\alpha\right)\left(z_1^\alpha - z_4^\alpha\right)}. \tag{51}$$

In bulk language this expression counts the contribution of a single graviton exchange between $V$ and $W$ in the background defined by the insertion of $O_H$. As observed in [57], this is enough to compute the chaos exponent.

We eventually want to extract the out-of-time-ordered correlator as in (49) by taking the Lorentzian continuation of Euclidean correlator on the $s$-plane. The analysis proceeds almost exactly as in [58], indipendently of whether $H$ is greater than or less than $c/24$. As we previously mentioned, in order to see the thermal nature of the eigenstate $H$, we need to place the $V$ and $W$ operators along the unit circle, that is $(z_i, \bar{z}_i) \rightarrow \left(e^{i x_i}, e^{-i \bar{x}_i}\right)$, and we will furthermore take

$$x_1 = t - i\epsilon_1 \,, \qquad\qquad \bar{x}_1 = -t + i\epsilon_1 \,, \tag{52}$$

$$x_2 = t - i\epsilon_2 \,, \qquad\qquad \bar{x}_2 = -t + i\epsilon_2 \,, \tag{53}$$

$$x_3 = x - i\epsilon_3 \,, \qquad\qquad \bar{x}_3 = x + i\epsilon_3 \,, \tag{54}$$

$$x_4 = x - i\epsilon_4 \,, \qquad\qquad \bar{x}_4 = x + i\epsilon_4 \,. \tag{55}$$

To obtain the time ordering as shown in (49), we need to ensure that the Lorentzian continuation starting from $t = 0$ is taken with

$$\epsilon_1 > \epsilon_3 > \epsilon_2 > \epsilon_4 \,. \tag{56}$$

This results in a Regge-limit of (50) whereby we go around the branch cut at $s = 1$ from *below*,[5] before taking $s \rightarrow 0$ as we take $t > x$. In the antiholomorphic sector, no branch cuts are crossed in the $\bar{s}$-plane and it remains small in the Lorentzian continuation. Multiplying holomorphic and antiholomorphic contributions in this limit gives:

$$G_{\text{Regge}} = \mathcal{F}_0(s)\bar{\mathcal{F}}_0(\bar{s}) = 1 - \frac{48\pi i \, h_V \, h_W}{c \, s} + \dots \,. \tag{57}$$

As was mentioned in the introduction, this form of the Regge correlator has a simple interpretation, namely that a single 'scramblon' with propagator $s^{-1}$ is exchanged between the operators in the OTOC. The effect of the heavy fields is to dress this propagator with a factor mimicking the thermal case, $s \sim e^{-i\alpha(t-x)}$, as depicted in Figure 1. In the end we find, for small $\epsilon_i$ (and defining $\epsilon_{ij} \equiv \epsilon_i - \epsilon_j$):

$$\text{OTOC} = 1 - \frac{192 i \pi \, h_V \, h_W/c}{\alpha^2 \epsilon_{12} \epsilon_{34}} \sin^2\left[\frac{\alpha}{2}(t - x)\right] + \dots \,. \tag{58}$$

Notice that for $H < c/24$ the OTOC oscillates and gives no evidence of scrambling. This is reminiscent of a CFT in a non-ergodic phase, which we discuss further in the next section.

As we increase the energy eigenvalue through $H > c/24$, we must take $\alpha \rightarrow i|\alpha|$ in (58) and we see that the large $t$ behavior exhibits the expected Lyapunov behavior:

$$\text{OTOC} = 1 - \frac{48 i \pi \, h_V \, h_W/c}{|\alpha|^2 \epsilon_{12} \epsilon_{34}} e^{|\alpha|(t-x)} + \dots \,, \tag{59}$$

with Lyapunov exponent

$$\lambda_L = |\alpha| = \frac{2\pi}{\beta_H} \,, \tag{60}$$

as promised. Note that the eigenstate butterfly velocity $v_B^H$ is the speed of light. Note also that the period of oscillation of the OTOC in the non-ergodic phase is governed by the parameter $\alpha \in \mathbb{R}$, which can be thought of as the analytic continuation of the Lyapunov exponent $\lambda_L$ to purely imaginary values.

---

[5] In [58] the branch cut is crossed from above in the Regge limit. Our analysis differs only because we are taking $\epsilon_1 > \epsilon_2$ rather than the reverse order.

It is important to mention that the authors of [59] also studied higher point correlators in heavy eigenstates and concluded that the leading answer, for an even number of light insertions, is given by a product of two-point functions of light operators in the heavy eigenstate, that is:

$$\langle O_H(\infty)\, \mathcal{Q}_1(x_1)\mathcal{Q}_2(x_2)\dots \mathcal{Q}_{n_L}(x_{n_L})\, O_H(0)\rangle \approx \prod_{ij}\langle O_H(\infty)\, \mathcal{Q}_i(x_i)\, \mathcal{Q}_j(x_j)\, O_H(0)\rangle + \text{perms}.$$
(61)

This is the leading disconnected piece of the correlator and is not in contradiction with our answer, as can be seen from (47). Our results have allowed us to extract the subleading, connected contribution.

# 5 Discussion: signatures of non-ergodic scrambling

One typical aim in holography is to match bulk calculations with analogous ones in CFT. States with operator dimensions below the threshold $H < c/24$ are dual to bulk geometries that are not black holes, but rather conical defects. This begs the question—what should we expect of these states? Since the bulk has no black hole, it comes as no surprise that these states fail to satisfy ETH in the same way that heavy operators do. In this section we will argue that CFT states dual to conical defects exhibit non-ergodic behavior, such as is typically encountered in many-body-localized (MBL) or spin glass phases.[6]

First consider the entanglement entropy of an interval of length $L$ computed in a eigenstate of dimension $H$, at large-$c$. This was first computed in [51, 62]:

$$S_{\text{EE}} = \frac{c}{3}\log\left[\frac{2}{\epsilon}\frac{R}{\alpha}\sin\left(\frac{L}{2R/\alpha}\right)\right],$$
(62)

where we have reinstated the size of the CFT circle $R \neq 1$. Above the $H > c/24$ threshold, we again identify $\alpha = 2\pi i R/\beta_H$ and the entanglement entropy behaves thermally. Below threshold the entanglement entropy obeys an area law as in vacuum, but where the CFT volume appears renormalized $R \to R/\alpha$. States where $S_{\text{EE}}$ obeys an area law were characterized as many-body-localized *states* in [63] and are meant to exhibit the physics of an MBL phase similarly to how pure states satisfying ETH exhibit physics in a thermal ensemble. This simple observation suggests that the below threshold states probe a non-ergodic phase of the CFT.

Further evidence for interpreting below threshold states as non-ergodic is provided in [64], in which the authors solve for the time-dependence of the entanglement entropy under a particular oscillatory type of driving. The dynamics can be solved because the driving force can be undone by an appropriate coordinate transformation. If this coordinate transformation is in the elliptic class, then the dynamics was dubbed 'non-heating,' in [64]. And for states with $H < c/24$, the coordinate transformation (28) is precisely in the elliptic class. Conversely when $H > c/24$ the coordinate transformation is in the hyperbolic class. When the dynamics fall in this class, [64] found that the entanglement entropy is thermalizing. Let us remark that the classification into elliptic, parabolic and hyperbolic also leaves signatures of (non-) ergodic behavior in late-time correlation functions [6, 39, 44, 65].

We finally come to the oscillating OTOC found in (58) when $H < c/24$. Because the OTOC does not decay, the immediate interpretation must be that these states mimic non-ergodic ensembles. To our knowledge, few examples of oscillating OTOCs similar to (58) have been computed. These include those computed numerically in [66] in a spin chain known to

---

[6]The idea that spacetime geometry may capture the physics non-ergodic systems through holography has been explored in [60, 61].

exhibit an MBL phase, giving further credence to the interpretation that these conical deficit states display the physics of non-ergodic systems. Another interesting example is the classically chaotic system known as the stadium billiard [67]. The eigenstate OTOCs were computed numerically and were found to be oscillatory [68], but interestingly, the stadium billiard's thermal OTOC, while non-oscillatory, also somehow fails to exhibit any Lyapunov behavior—a stark difference between its classical and quantum behavior, although perhaps we can attribute this to the billiard's small number of degrees of freedom. This illustrates an important and more general point, namely that thermalizing systems do not necessarily scramble efficiently. A striking example, taking us again closer to the black-hole context, is the IOP matrix model [69, 70], which shows signatures of thermalization and information loss, yet does not have an exponential OTOC, much less one that saturates the bound [71]. It will be interesting to further investigate non-ergodic OTOCs in holographic models as well as more generic chaotic quantum systems [72, 73].

Curiously, an oscillatory OTOC at finite temperature was observed in [74]. This OTOC was computed holographically in a geometry that interpolates between an AdS$_2$ boundary and a de Sitter horizon deep in its interior [75], suggesting that de Sitter horizons may themselves be non-ergodic, despite being at finite temperature.

## Acknowledgments

We thank Tom Hartman for collaboration on related issues, and for numerous discussions on the topic of this work. We would also like to thank Dionysios Anninos, Alexandre Belin, Shouvik Datta, Manuela Kulaxizi, Kyriakos Papadodimas, Andrei Parnachev, and Alexander Zhiboedov for discussions. T.A. is supported by the Natural Sciences and Engineering Research Council of Canada, by grant 376206 from the Simons Foundation, and by the National Science Foundation Grant No. NSF PHY-1748958. T.A. also acknowledges the hospitality of the KITP where part of this work was completed. This work has been supported by the Fonds National Suisse de la Recherche Scientifique (Schweizerischer Nationalfonds zur Förderung der wissenschaftlichen Forschung) through Project Grants 200021_162796 and 200020_182513 as well as the NCCR 51NF40-141869 "The Mathematics of Physics" (SwissMAP).

## A   Eigenstates with spin

In the main text we have been careful to consider eigenstates with equal left- and right-moving conformal dimensions when defining the microcanonical temperature, but this can be generalized to the case where the operator has spin, with minimal change. For spinning particles ($H \neq \bar{H}$), the density of states at high energy ($\sqrt{E_L E_R} > c/24$) is given by the Cardy formula:

$$S(E_L, E_R) = 2\pi \sqrt{\frac{c}{6} E_L} + 2\pi \sqrt{\frac{c}{6} E_R} \, , \qquad (63)$$

with

$$E_L \equiv H - \frac{c}{24} \, , \qquad E_R \equiv \bar{H} - \frac{c}{24} \, . \qquad (64)$$

We can thus define two distinct microcanonical temperatures that left- and right-movers will be sensitive to:

$$\beta_H \equiv \partial_{E_L} S(E_L, E_R) = \frac{2\pi}{\sqrt{\frac{24H}{c} - 1}} \, , \qquad \beta_{\bar{H}} \equiv \partial_{E_R} S(E_L, E_R) = \frac{2\pi}{\sqrt{\frac{24\bar{H}}{c} - 1}} \, . \qquad (65)$$

Note that $\beta_H$ defined in this way coincides with (9), as it should.

# B    Non-vacuum HHLL block

In the main part of this paper we have shown how straightforward it is to compute the vacuum HHLL block using this trick, we now show how to compute the non-vaccuum blocks. We will see that they require more subtle considerations.

Remember that to set up this problem, we first fixed as many accessory parameters as possible using the regularity conditions (18). In turn, this means that we cannot use regularity in the $w$ plane to fix even more accessory parameters. The rest must be obtained by solving the monodromy problem. So, while it is perfectly fine to assume that $T_{\mathrm{cl}}(w)$ should fall off like $w^{-4}$ in the vacuum block, it is not fine to assume this whenever there is non-vacuum exchange. The reason it works for the identity block stems from the fact that the stress tensor obeys regularity at infinity *in this block only*, due to all operators fusing to the identity.

To solve the problem in the $w$ plane, we will use the method of variation of parameters. The idea is to use the fact that $h_L/c \equiv \varepsilon \ll 1$ such that

$$\psi = \chi + \varepsilon\,\eta + \dots \tag{66}$$

$$T_{\mathrm{cl}} = \varepsilon\,T_L \ . \tag{67}$$

To zeroth order in $\varepsilon$, the independent solutions are $\chi = (1, w)$. It is now standard to find $\eta$:

$$\eta_i = \left\{ \int^w F_i{}^j \right\} \chi_j \ , \qquad F_i{}^j \equiv \frac{\chi_i\,\epsilon^{jk}\chi_k}{\chi_1\chi_2' - \chi_2\chi_1'} T_L \ . \tag{68}$$

This in turn allows us to write down the monodromy matrix to first order in $\varepsilon$ around a path $\gamma$ that encircles the points 1 and $w_2$:

$$M_\gamma = 1_{2\times 2} + 2\pi i\varepsilon \left( \mathrm{Res}_{w=1} F + \mathrm{Res}_{w=w_2} F \right) \ . \tag{69}$$

The eigenvalues of the full monodromy matrix should be:

$$\mathrm{eigenvalues}(M_\gamma) = \exp\left\{ i\pi \left( 1 \pm \sqrt{1 - \frac{24 h_p}{c}} \right) \right\} \ , \tag{70}$$

meaning that we can match the leading order expression in $h_p/c$ between (69) and (70) yielding:

$$\tilde{c}_2 = -\frac{6}{c} \frac{\left(2 h_L - h_p\sqrt{w_1/w_2}\right)}{w_1 - w_2} \ . \tag{71}$$

This agrees with (37) when $h_p \to 0$. Also, now with hindsight to guide us, we see that we could have obtained this answer by demanding

$$\tilde{T}_{\mathrm{cl}}(w \to \infty) = -\frac{6 h_p/c}{w\sqrt{w_1 w_2}} + O(w^{-2}) \ . \tag{72}$$

We can now find $\tilde{f}(w_1, w_2)$ by solving

$$\frac{\partial \tilde{f}(w_1, w_2)}{\partial w_2} = \tilde{c}_2 \ , \qquad \frac{\partial \tilde{f}(w_1, w_2)}{\partial w_1} = \frac{6}{c\,w_1}\left( 2 h_L - \frac{c}{6}\tilde{c}_2\,w_2 \right) \ , \tag{73}$$

which can be read off from the expression for $\tilde{c}_1$ in (31). We can fix the constant of integration partly by demanding

$$\tilde{f}_p(w_2 \to w_1) \sim \frac{6(2 h_L - h_p)}{c} \log(w_1 - w_2) \ . \tag{74}$$

This gives

$$\tilde{f}_p(w_1, w_2) = \frac{12h_L}{c} \log(w_1 - w_2) - \frac{6h_p}{c} \left[ \log(w_1 - w_2) - 2\log\left(\frac{\sqrt{w_1} + \sqrt{w_2}}{2}\right) \right] + \text{const} . \quad (75)$$

Now comes the important observation. If we use (36) to go from this expression to expression for the semiclassical block on the $z$ plane, we get the wrong answer. Instead we must add an additional 'constant' of integration proportional to $h_p$:[7]

$$f_p(z_i) = \frac{12H}{c} \log(y_1 - y_2) + \tilde{f}_p(w(z_i)) - \frac{6}{c} \left[ h_L \log w'(z_2) + h_L \log w'(z_1) \right] - \frac{6h_p}{c} \log\left[ \frac{z_1 - z_2}{\alpha(1-z)} \right] \quad (76)$$

$$= \frac{12H}{c} \log(y_1 - y_2) + \frac{6}{c} \left\{ 2h_L \log\left[ (z_1 - z_2) \frac{z^{\frac{1-\alpha}{2}}(1-z^\alpha)}{\alpha(1-z)} \right] \right.$$
$$\left. - h_p \log\left[ \frac{4}{\alpha} \frac{z_1 - z_2}{1-z} \left( \frac{1-z^{\frac{\alpha}{2}}}{1+z^{\frac{\alpha}{2}}} \right) \right] \right\} , \quad (77)$$

yielding the correct HHLL block, with the cross ratio

$$z \equiv \frac{(z_1 - y_2)(z_2 - y_1)}{(z_1 - y_1)(z_2 - y_2)} . \quad (78)$$

The 'integration constant' proportional to $h_p$ in (76) comes from the need to impose the correct OPE singularity on the $z$ plane. For standard insertion locations $(y_1, y_2, z_1) = (0, \infty, 1)$ it is an actual constant, but shows that the non-identity block cannot simply be obtained by coordinate transforming the solution to the $w$-plane problem. More work needs to be done.

We hope that this example is instructive, as it contains one of the messages of our paper. In 2d CFT at large-$c$, the heavy insertions do indeed behave as a coordinate transformation, but the need to impose the correct OPE singularity on the $z$ plane implies that each individual conformal blocks, apart from the identity block, transforms anomalously under the coordinate transformation. However, as previously noted, these complications do not arise in the vacuum block, meaning that the vacuum block is the one that truly appears thermal in CFT$_2$.

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
