# Peer review of "Phases of scrambling in eigenstates"

_SciPost Physics, doi:SciPost Phys. 7, 003 (2019)_

## Round 2 · Referee Report · Anonymous (Referee 1) · 2019-6-4

Strengths

1- Well written, transparent structure 2- Necessary background is explained in a useful fashion 3- Presentation of new results is intuitive, but at the same time precise 4- Content is interesting and timely 5- Referencing is good

Weaknesses

1- The content is interesting and useful, though perhaps a little bit limited in scope

Report

This paper investigates a version of the eigenstate thermalisation hypothesis in the context of 2d CFTs with large central charge and a sparse spectrum. The authors show that correlation functions of any even number of (simple) probe operators in heavy primary states look approximately like thermal expectation values. The relevant temperature is the ETH temperature associated with the heavy state. As an application, the authors compute the OTOC of four probe operators in a heavy primary state and show that under the above assumptions it exhibits maximal Lyapunov growth with the Lyapunov exponent given by the same microcanonical temperature associated with the heavy state. Interestingly, there is a sharp transition between exponential Lyapunov growth and oscillatory behaviour when the heavy primary creating the state approaches the BTZ threshold.

The paper is very well written, clearly structured, and shows appropriate awareness of previous literature. The authors explain all the necessary background in a pedagogical fashion. The paper definitively adds valuable information to the current state of the field. I will recommend the paper for publication.

One comment though, regarding identity block dominance: I understand that the full implications/justifications for this assumption are probably not entirely understood even for the HHLL block. But I wonder if the authors have some more to say about this assumption in the case of intermediate channels between light operators in the HHLL...L blocks. In particular, does the method presented here require a hierarchy of conformal weights of the string of light operators ($h_{Q_1} \gg h_{Q_2} \gg \ldots$)? If so, would one hope for the results about OTOCs to hold outside this regime?

typos: - Caption of figure 2. - First sentence on page 14.

---

## Round 3 · Author Response

Dear Editor,

We have addressed the referee’s comments and fixed the two typos they pointed out. We have also seized the opportunity to correct two further minor typos. See list of changes.

Reply to referee:

We would like to thank the referee for their careful reading and insightful report. We reply to their comment as follows:

The referee raises the important point of the validity of the assumption of identity block domination and whether a further hierarchy among the light operators is necessary. Such an additional hierarchy is not necessary for our result to hold, one merely needs to assume that the total contribution to the semiclassical stress tensor coming from the aggregate effect of the Q operators is $O(\varepsilon c)$ in the notation of our paper.

As an example we have shown how to extract the Lyapunov exponent from a six-point function between two heavy and four light operators, with no additional hierarchy between the latter. Nevertheless it would be interesting to investigate if further mileage can be gained from making the additional assumption of such a hierarchy.

---

## Round 3 · List of Changes

igure 2: replaced erroneous ‘=' sign with the word ‘operators’.
Page 8, below Eq. (3.10): fixed an incorrect subscript on the matrix $M$.
Top of page 14: we fixed the sentence structure, as pointed out by the referee.
Page 17: minor typo fixed (“does not decays” replaced by “does not decay”).

---

## Editorial Decision

published